# Chalcone-Synthase-Encoding *RdCHS1* Is Involved in Flavonoid Biosynthesis in *Rhododendron delavayi*

**DOI:** 10.3390/molecules29081822

**Published:** 2024-04-17

**Authors:** Ju Huang, Xin Zhao, Yan Zhang, Yao Chen, Ximin Zhang, Yin Yi, Zhigang Ju, Wei Sun

**Affiliations:** 1Key Laboratory of State Forestry Administration on Biodiversity Conservation in Karst Mountain Area of Southwest of China, School of Life Science, Guizhou Normal University, Guiyang 550025, China; 15285737718@163.com (J.H.); zxjyn536@163.com (X.Z.); 15186346268@163.com (Y.Z.); 18984245819@163.com (Y.C.); zhxm409@163.com (X.Z.); yiyin@gznu.edu.cn (Y.Y.); 2Pharmacy College, Guizhou University of Traditional Chinese Medicine, Guiyang 550025, China

**Keywords:** flower color, flavonoid, chalcone synthase, enzyme activity, *Rhododendron delavayi*

## Abstract

Flower color is an important ornamental feature that is often modulated by the contents of flavonoids. Chalcone synthase is the first key enzyme in the biosynthesis of flavonoids, but little is known about the role of *R. delavayi* CHS in flavonoid biosynthesis. In this paper, three *CHS* genes (*RdCHS1-3*) were successfully cloned from *R. delavayi* flowers. According to multiple sequence alignment and a phylogenetic analysis, only RdCHS1 contained all the highly conserved and important residues, which was classified into the cluster of bona fide CHSs. RdCHS1 was then subjected to further functional analysis. Real-time PCR analysis revealed that the transcripts of *RdCHS1* were the highest in the leaves and lowest in the roots; this did not match the anthocyanin accumulation patterns during flower development. Biochemical characterization displayed that RdCHS1 could catalyze *p*-coumaroyl-CoA and malonyl-CoA molecules to produce naringenin chalcone. The physiological function of *RdCHS1* was checked in *Arabidopsis* mutants and tobacco, and the results showed that *RdCHS1* transgenes could recover the color phenotypes of the *tt4* mutant and caused the tobacco flower color to change from pink to dark pink through modulating the expressions of endogenous structural and regulatory genes in the tobacco. All these results demonstrate that RdCHS1 fulfills the function of a bona fide CHS and contributes to flavonoid biosynthesis in *R. delavayi*.

## 1. Introduction

Flower color, a vital trait of ornamental plants, is mainly determined by flavonoids [1,2]. Flavonoids are a large group of plant natural pigments that comprise chalcones, aurones, flavones, isoflavones, flavandiols, flavonols, proanthocyanins, anthocyanins, and so on [3]. In addition to tissue pigmentation, flavonoids also fulfill significant roles in multiple physiological processes, such as UV protection, auxin transport, phytopathogens, defense against herbivores, signaling between plants and microbes, and pollen development [4,5]. Importantly, flavonoids provide a wide variety of health benefits to humans, including delaying the aging of immune organs and the nervous system, eyesight improvement, and the prevention of cancer, Alzheimer’s disease, and cardiovascular disease [6,7]. Extensive studies on flavonoid biosynthetic pathways have been completed using *Petunia hybrida* (petunia) [8], *Antirrhinum majus* (snapdragon) [9], and *Arabidopsis thaliana* (Arabidopsis) [10] as models, and many enzymes that take part in flavonoid biosynthesis have been characterized [11].

Chalcone synthase (CHS), a polyketide synthase, is crucial in the biosynthetic pathway of flavonoids and serves as the gatekeeper to regulate their biosynthesis [12]. CHS catalyzes a three-step condensative reaction from trimolecular malonyl-CoA and monomolecular *p*-coumaroyl-CoA to produce naringenin chalcone [13]. For the first step, the coumaroyl moiety is loaded to the site of active cysteine (Cys164). Next, condensation reactions occur based on the decarboxylation of malonyl-CoA to produce nucleophiles for chain elongation. Finally, the reaction generates thioester-linked tetraketide, which cyclizes into a hydroxylated aromatic ring to yield chalcone [14]. Due to their important roles in initiating flavonoid biosynthesis, CHSs have been intensively researched in many higher plants, including petunia [8], *Arabidopsis* [10], *Antirrhinum* sp. [9], *Oncidium orchid* [15], apple [16], grape [17], *Gerbera hybrida* [18], *Dianthus chinensis* [19], and so on. In *Antirrhinum majus*, the first *CHS* mutant was called white nivea, generated through deleting its single *CHS* gene [20]. For petunia, there are more than eight copies of *CHS* genes in its genome, but only the *CHS-A* and *CHS-J* transcripts are expressed in its floral tissues. They are conspicuously down-regulated in the white parts of ‘Red Star’ [21,22]. In addition, the spatial suppression of *CHSA* also causes the natural bicolor floral phenotype to be affected, as well as an inability to generate functional pollen tubes [23]. In contrast to petunia, maize *CHS* mutants can still initiate pollen tube growth but are incapable of sustaining growth exceeding 12 h, ultimately leading to male sterility [24]. The *Arabidopsis tt4* mutant, which lacks brown tannins in its seed and anthocyanin in the cotyledons and hypocotyls, has been demonstrated to be caused by a mutation in the *CHS* gene. Other phenotypic effects, such as changes in pollen fluorescence and root morphology, were also reported later [25,26]. In *Gerbera hybrida*, three *CHS*-like genes are expressed during its flower development, whereas only the *CHS1* transcript corresponds with the synthesis of anthocyanins and flavanols [27]. As for the *Asiatic hybrid* lily, there are also three *CHS* genes (*CHSA-CHSC*) that are active in colored tepals, but their transcript patterns are diverse [28,29]. Therefore, all the mentioned results indicate that CHS plays an indispensable role during plant development and is crucial for petal color formation in some plants.

*Rhododendron delavayi* (*R. delavayi*), which belongs to the Ericaceae family, is a significant ornamental plant species. Considering its vivid flowers and high horticultural value, *R. delavayi* has become increasingly popular in the world [30]. However, the first rate-limiting enzyme, chalcone synthase, which is vital for petal coloration, has not been isolated and analyzed from *R. delavayi*. For this study, a *CHS* gene (named *RdCHS1*) was isolated from the petals of *R. delavayi*, and its functional roles in flavonoid biosynthesis were comprehensively demonstrated. An expression analysis of *RdCHS1* was conducted, showing that it might participate in the biosynthesis of flavonoids (not only anthocyanin) in all detected tissues. Subsequently, the catalytic property of RdCHS1 was confirmed, and its function in planta was verified through transferring it into the *Arabidopsis tt4* mutant. At the same time, *RdCHS1* was also ectopically expressed in tobacco; the data displayed that it could strengthen the pink color of the corolla from pale pink to dark pink. In this paper, we comprehensively identify the function of chalcone synthase in *R. delavayi* and prove that *RdCHS1* is a *CHS* gene affecting flavonoid biosynthesis in *R. delavayi*.

## 2. Results

### 2.1. Cloning and Sequence Analysis of RdCHSs

According to the transcriptome data of *R. delavayi*, three *CHS* genes were identified through blastn alignment, with reference genes from *Arabidopsis* and proximal species. The cDNA sequences of *RdCHS1* (SUB13889370), *RdCHS2,* and *RdCHS3* were cloned and found to have the ORFs of 1182 bp, 762 bp, and 1170 bp, encoding 393-, 253-, and 389-amino acid proteins, respectively. Protein sequence alignment for the RdCHSs protein (in comparison to other plant CHSs) was conducted; the results disclosed that RdCHS1, RdCHS2, and RdCHS3 all contained three highly conserved residues (marked with a yellow box), which form the active site of CHS. However, the second important Phe residue (marked with a blue box) determining the catalytic specificity of CHS was replaced with Val in RdCHS2 and RdCHS3. Meanwhile, the ORFs of *RdCHS2* were significantly shorter than the typical lengths of CHSs, leading to a deletion of the first five residues (marked with a black triangle) that help form the coumaroyl-CoA binding and cyclization pocket. In two signature motifs (marked with a green box), only RdCHS2 and RdCHS3 contained different amino acid residues compared to other CHSs, the function of which was identified (Figure 1). Moreover, the phylogenetic analysis also revealed that RdCHS1 was grouped into the cluster of bona fide CHSs and is evolutionarily closest to the CHS from *Vitis vinifera*, while RdCHS2 and RdCHS3 were classified into non-CHS Type III PKS clusters. Thus, based on the unusual ORF lengths and variations in the conserved residues and motifs, as well as the phylogenetic analysis, RdCHS1 was subjected to further functional analysis (Figure 2).

### 2.2. RdCHS1 Expression Patterns in Developing Flowers and Different Tissues

The transcript patterns of RdCHS1 were checked in *R. delavayi* using real-time PCR. RdCHS1 was expressed globally in all tissues and varied according to tissue type, with the highest transcript abundance detected in the leaves and the lowest in the roots. Furthermore, similar levels of RdCHS1 mRNA were observed in flower organs, including petals, toruses, scapes, pistils, and stamens (Figure 3A). Then, more detailed transcript patterns of RdCHS1 were examined during different flower developmental stages. As shown in Figure 3B, the expression profile of RdCHS1 increased steadily in the early two stages, decreased in the following two stages, and achieved its highest expression level in the last stage, which is inconsistent with the anthocyanin synthesis patterns during R. delavayi flower development [31]. Therefore, the above-mentioned findings suggest that RdCHS1 may be involved in the biosynthesis of not only anthocyanin, but also the other flavonoids in the detected tissues.

### 2.3. Biochemical Characterization of RdCHS1

As a crucial step in confirming the biological function of RdCHS1, it was heterologously expressed as a thioredoxin-fusion protein along with His-tag in *E. coli*. The recombinant purified RdCHS1 displayed a single protein band in SDS-PAGE (Figure 4A). Then, in vitro enzymatic assays were performed to elucidate the catalytic activity of the recombinant RdCHS1 in the presence of malonyl-CoA and *p*-coumaroyl-CoA. As reported previously, naringenin chalcone (NC), the catalytic product of CHS, can spontaneously convert into naringenin in aqueous solutions [32]. Therefore, both naringenin chalcone and naringenin were regarded as the products of recombinant RdCHS1 in an HPLC analysis. Compared to the control, naringenin chalcone and naringenin products were detected in reactions with the RdCHS1 protein, which matched the authentic samples (Figure 4B–D). This result indicates that RdCHS1 encodes a biochemically functional CHS protein, catalyzing the synthesis of naringenin chalcone from malonyl-CoA and *p*-coumaroyl-CoA.

### 2.4. Complementation of the tt4 Mutant with RdCHS1

In order to determine the effect of *RdCHS1* on anthocyanin biosynthesis, it was transformed into the *Arabidopsis tt4* mutant, and more than 10 independent lines were obtained. In comparison, wild-type *Arabidopsis* accumulated anthocyanins in their hypocotyls and tannins in their seed coats, but the *tt4* mutant could not. As presented in Figure 5A, transformation with *RdCHS1* could successfully recover the pigmentation phenotype of the *tt4* mutant. Meanwhile, RT-PCR was also carried out to affirm the expressions of *RdCHS1*, and the amplicons absent in Col-0 and the *tt4* mutant were observed in the transgenic seedlings (Figure 5B). Additionally, to determine the change in anthocyanins in detail, an HPLC analysis was conducted. The results in Figure 6 show that the anthocyanins in the wild-type *Arabidopsis* (peaks 1–4) were all absent in the *tt4* mutant, while the seedlings carrying *RdCHS1* could successfully complement the biosynthesis of cyanindin and pelargonidin, which coincided with the quantification of total anthocyanins (Appendix A, Figure 5C). Therefore, the above results imply that RdCHS1 possesses similar activity to AtCHS, a chalcone synthase involved in flavonoid biosynthesis in vivo.

### 2.5. Overexpression of RdCHS1 in Tobacco

In order to verify the function of *RdCHS1* in the biosynthesis of anthocyanin in flowers, it was synchronously transformed into tobacco plants. After screening, 15 independent transgenic lines were generated, and two independent lines exhibiting a significantly strengthened flower color were selected for further analysis (Figure 7A). The presence of *RdCHS1* on a molecular level was examined using RT-PCR, and the contents of anthocyanin in tobacco corollas were also determined through HPLC (Figure 7B,D). The quantitative results displayed that the amount of anthocyanin in the transgenic flowers was higher than that in the non-transformed flowers, which accounted for 131.7–184.2% of the total anthocyanins in the control (Figure 7C). Thus, it became clear that the ectopic expression of *RdCHS1* can increase anthocyanin accumulation in tobacco flowers.

The increased anthocyanin in the transgenic tobacco flowers implied that a coordinated interaction might exist between *RdCHS1* and the endogenous enzymes involved in the biosynthesis of anthocyanin. Hence, real-time PCR analysis was carried out to investigate the effect of *RdCHS1* over-expression on the anthocyanin pathway in the transgenic flowers. As presented in Figure 8, all the investigated genes except *NtF3′5′H* were consistently up-regulated compared to in wild-type tobacco. Moreover, the ectopic expression of *RdCHS1* in the tobacco strongly influenced the transcript levels of *NtCHS* and *NtAN2*, and their transcript abundances were from 2.4 to 5.51-fold higher in both transgenic lines. Thus, these findings reveal that *RdCHS1* overexpression can modulate the expressions of the endogenous anthocyanin pathway genes in tobacco.

## 3. Discussion

The CHS enzyme family is crucial for plant growth and development. They are ubiquitous in different plant species and encoded by multiple genes. For example, 3, 4, 12, and 14 *CHS* genes have been cloned and identified from *Lilium* spp. [28,29], *Dahlia variabilis* [33], *Zea mays* [34], and *Petunia hybrida* [35], with several of them being true CHSs, whereas others are involved in various metabolic pathways [36]. *Arabidopsis* has four *CHS* genes, one of which is the true CHS and takes part in flavonoid biosynthesis [37]. Of the RdCHSs, the protein sequence alignment and phylogenetic analysis showed that RdCHS1 was closely related to the bona fide CHSs, indicating its ability to produce naringenin chalcone (Figure 1 and Figure 2).

CHSs in plants share a high similarity in their amino acid sequences. The RdCHS1 amino acids were 82.6–91.4% identical with the *Arabidopsis* CHSs and *Vitis vinifera* CHSs (Figure 1). An increasing number of PKSs (except for CHS), such as stilbene synthase (STS), acridone synthase (ACS), and 2-pyrone synthase (2PS), have been demonstrated to possess an identical catalytic mechanism to that of CHSs, but they are different in terms of their intramolecular cyclization patterns and predilection for starter substrates [38,39]. The above enzymes are also quite similar at the amino acid level with CHSs. Thus, plentiful CHS sequences in public databases, identified only through their sequence similarity, may, in fact, encode other related enzymes [40]. Alternatively, the kinds of amino acids in protein sequences might help to define the CHS. The Phe residues (Phe215 and Phe265) are two gatekeepers in CHSs that facilitate substrate loading and appropriate folding during the cyclization process [14,38]. Phe265, a critical residue for substrate selectivity, is conserved in the sequence of CHSs, but changes in other plant PKSs [39,41]. In OsCHS9, Phe265 is substituted with Gly, which displays undetectable CHS activity; instead, it encodes GUS, catalyzing the production of bisdemethoxycurcumin [42]. RdCHS1 contains both Phe residues in its amino acid sequence, implying that it may be a functional CHS in *R. delavayi*, and exhibits CHS activity. In RdCHS2 and RdCHS3, Phe265 is replaced with Val, indicating their function as other plant PKSs (Figure 1); thus, RdCHS1 was selected for further analysis.

Gene expression patterns are correlated with their functions; a differential analysis of gene expression can provide key information for the study of gene features, regulation, and origins [43,44]. The expression patterns of *RdCHS1* are spatially regulated. Its highest mRNA level was detected in leaves, followed by flower tissues, and it was the lowest in roots. Meanwhile, *RdCHS1* expression in flowers is also developmentally regulated, but it is not associated with the accumulation of total anthocyanins (Figure 3). These are different from the results in *Petunia hybrida* and *Gerbera hybrida*, in which CHS expression is excessive in flowers and is coupled with anthocyanin pigmentation [21,27]. Together with its high expression in leaves, the *RdCHS1* expression profiling in this research suggests that it might not be the dominant CHS enzyme in *R. delavayi* petals.

In vitro enzymatic assays showed that RdCHS1 was an authentic CHS similar to the CHS enzymes in *Gerbera hybrida* (GCHS1 and GCHS3) that could convert malonyl-CoA and *p*-coumaroyl-CoA molecules into naringenin chalcone (Figure 4) [18]. Although RdCHS1 performed typical CHS functioning, it did not seem to be a major functional CHS in petal pigmentation, according to its highest expression in the leaves and lower catalytic efficiency towards malonyl-CoA and *p*-coumaroyl-CoA. Recent studies have reported that the CHS gene in *Physcomitrella patens* can accept dihydro-*p*-coumaroyl-CoA and cinnamoyl-CoA to produce relevant chalcones [45]. Similarly, the *CHS* gene from *Scutellaria baicalensis* also has the ability to convert isovaleryl-CoA, phenylacetyl-CoA, isobutyryl-CoA, and benzoyl-CoA into a variety of products containing the aromatic polyketide, which is unnatural [46]. Therefore, CHS is generally a promiscuous enzyme in regard to substrate specificity, which suggests its functional diversification during the process of evolution. In view of the above discoveries, further research is needed to explore the catalytic properties of RdCHS1, so as to determine its function in plants and also lay the foundation for functional divergence studies of the *CHS* gene family in *R. delavayi*.

A few studies have observed the influence of reducing CHS activity in various plants, such as in *Arabidopsis*, where *tt4* was the CHS gene mutant and displayed a deficiency in the synthesis of anthocyanin, as well as an absence in the pigment of the seed coating [47]. Thus, the *tt4* mutant is a suitable model for verifying whether *RdCHS1* takes part in anthocyanin and proanthocyanidin biosynthesis. As expected, *RdCHS1* completely restored the purple coloration of *tt4* in hypocotyls and cotyledons and the pigment in the seed coats, which confirmed the function of RdCHS1 as a CHS in vivo (Appendix A, Figure 5). These results are similar to the case of maize; its *C2* gene encoding CHS was also overexpressed in *tt4* mutants and exhibited a similar pigmentation phenotype and accumulation patterns of flavonoids [48]. Meanwhile, the complementation of *Arabidopsis* flavonoid mutants was also conducted by other enzymes involved in flavonoid biosynthesis, such as chalcone isomerase from *Ophiorrhiza japonica* and dihydroflavonol 4-reductase from *Dryopteris erythrosora*; all these findings demonstrate that the function of enzymes that participate in flavonoid biosynthesis is exchangeable among different plant species [49,50].

When compared to wild-type tobacco, the overexpression of the *RdCHS1* gene resulted in dark pink flowers and increased cyanidin-type anthocyanins in transgenic lines (Appendix A, Figure 7), and a similar phenomenon has also been observed in *Solanum lycopersicum* [51]. Although enhancive expressions of endogenous structure genes (*NtCHS*, *NtCHI*, *NtF3H*, *NtF3′H*, *NtANS,* and *NtUFGT*) and regulatory factors (*NtAN1a*, *NtAN1b*, and *NtAN2*) (Figure 8) have been detected in *RdCHS1* transgenic tobacco (Figure 8), it is not clear whether the enhanced biosynthesis of anthocyanins is due to the induction of the endogenous pathway or the catalytic function of RdCHS1. Because a massive expression of *RdCHS1* can increase the metabolic flux of the anthocyanin pathway, it can, thus, promote anthocyanin accumulation. On the other hand, the up-regulation of these endogenous gene expressions may be attributed to the positive feedback regulation of flavonoid pathway intermediates such as naringenin chalcone (the product of RdCHS1), or that the RdCHS1 protein may directly interact with structural or transcriptional regulation proteins to increase anthocyanin accumulation [52]. Unexpectedly, the transcript level of *NtF3′5′H* remained constant, which may have been caused by the host tobacco being incapable of producing delphinidin-type anthocyanins due to dihydromyricetin deficiency [53]. Overall, the above results prove that *RdCHS1* encodes a biochemically functional CHS protein and takes part in flavonoid biosynthesis.

## 4. Materials and Methods

### 4.1. Plant Materials

*R. delavayi* was grown in the experimental field of Gui Zhou Normal University. The anthesis, scapes, petals, pistils, toruses, roots, stamens, leaves, and flowers at different stages (stages 1–5) were sampled. *Arabidopsis* T-DNA insertion mutant (*tt4*, SALK020583) and wild-type in Columbia ecotype background were obtained from NASC ( ) and cultivated in the glasshouse. T2 transgenic *Arabidopsis* seedlings cultured for 7 days were harvested from anthocyanin inductive medium (1/2 MS adding 3% sucrose) and used for RT-PCR and anthocyanin analysis. The tobacco (K326) plants used in the transformation were kept in 12 h of light at 22 °C. The blooming flowers of the T1 transgenic tobacco were collected. All the above plant samples were quick-frozen immediately and stored at −80 °C for later analysis.

### 4.2. Chemicals

Malonyl-CoA, *p*-Coumaroyl-CoA, and Naringin chalcone used in the enzyme activity analysis were obtained from Sigma-Aldrich (Saint Louis, MI, USA). Cyanidin 3-*O*-glucoside was purchased from Phytolab (Bayern, Germany).

### 4.3. Full-Length cDNA Cloning of RdCHS1

Extracted RNA from the flowers of *R. delavayi* was used for the synthesis of cDNA using Takara M-MLV reverse transcriptase. Based on the assembled transcriptomic information (SRR26283938), specific primers for *RdCHS* CDS (coding sequence) cloning were designed (Appendix A). After amplification, the products of the PCR were inserted into pMD18-T vectors (Takara, Japan) and transformed into the competent cells of JM109. After enzyme digestion verification, multiple positive clones were selected and subjected to sequencing to verify the accurate nucleic acid order of RdCHSs.

### 4.4. Sequence Alignment and Phylogenetic Analysis

The multi-alignment of different CHS sequences was performed with DNAMAN 5.0. The phylogenetic tree was built using CHS proteins from diverse plants in MEGA 6.0 with 2000 bootstrap replicates for estimating the confidence of the tree clade.

### 4.5. Gene Expression Analysis

The total RNAs were extracted from *R. delavayi* and tobacco samples. The BioRad CFX96 real-time PCR system was selected for the gene expression analysis with gene-specific primers (Appendix A). *RdActin* and *NtTublin* were selected as internal controls for the *R. delavayi* and tobacco samples, respectively. The PCR conditions were the same as those reported in a previous paper [31]. Each sample was run in triplicate, and the 2^−ΔΔCT^ method was used to calculate the expression values. To confirm the specific amplification, agarose gel electrophoresis and a melting curve analysis were carried out.

### 4.6. Soluble Protein Extraction and CHS Enzyme Assay

The complete ORF of *RdCHS1* was amplified using the PCR method. The resulting fragments were introduced into the *BamH* I and *Hind* III sites of the pET-32a vector and transformed into *Escherichia. coli* strain BL-21 to express the protein with an N-terminal His6 tag. Protein expression and purification were conducted, as reported previously [54]. Briefly, bacterial fluid containing the corresponding plasmid was induced for 48 h at 15 °C after adding IPTG. The recombinant RdCHS1 protein with the His tag was purified using the TransGen purification kit following the instructions. The protein purity was then analyzed through SDS-PAGE.

The chalcone synthase activity for generating naringenin chalcone was checked in the reaction mixtures, consisting of 160 μM malonyl-CoA, 80 µM *p*-coumaroyl-CoA, 100 mM potassium phosphate (pH 7.2), and 30 μg purified recombinant protein. After incubation at 30 °C for 60 min, the assay mixtures were terminated via extracting twice with 100 µL ethyl acetate followed by centrifugation for 20 min. The formed products were detected through high-performance liquid chromatography (HPLC) using a C18 column by monitoring the absorbance at 304 nm. The mobile phase was the miscible liquids of 3% acetic acid, 47% water, and 50% methanol.

### 4.7. Expression Vector Construction and Transformation of Arabidopsis and Tobacco

The pBI121 that contained the CaMV35S promoter and NPTII was used to construct a binary vector for *Arabidopsis* and tobacco transformation. The full-length CDS of *RdCHS1* was amplified from the pMD18-T vector and cloned into the pBI121 vector digested with *Xba* I and *BamH* I. After construction, the over-expression cassette was sequenced to confirm the correct insertions of *RdCHS1,* and then this was immobilized into the *Agrobacterium tumefaciens* strain GV3101 via the freeze–thaw method. *Arabidopsis* transformation was conducted according to the method employed in previous reports [55]. After sterilization, transgenic *Arabidopsis* seed selection was carried out on 1/2 MS medium supplemented with kanamycin (50 mg L^−1^). Following 1 week of growth, T2 *Arabidopsis* transgenic seedlings with purple cotyledons were sampled and used for later analysis. Meanwhile, tobacco transformation was also carried out following a previously reported protocol [56]. The transgenic tobacco seedlings were selected and grown in a greenhouse. Only the fully expanded flowers were harvested, they were used for further analysis. In order to confirm the over-expression of *RdCHS1*, RT-PCR analysis was performed in *Arabidopsis* and tobacco using *β-actin* and *NtTublin* as controls.

### 4.8. Anthocyanin Analysis of Transgenic Arabidopsis Seedlings and Tobacco Flowers

For the anthocyanin analysis, 0.2 g of freeze-dried samples from the transgenic *Arabidopsis* seedlings and tobacco flowers was powdered and extracted using 1 mL extracting solutions (H_2_O:MeOH:HCl = 75/24/1) for 14 h at 4 °C. The extract solution was centrifuged and filtered through 0.22 µm microporous filters. Then, the extracted compounds were detected with HPLC using a C18 column. Two eluents, 5% formic acid (A) and 1.5% methanol (B), were used for elution. Gradient elution was conducted at a 1.0 mL min^−1^ flow rate: 0–10 min, 86–83% A; 10–35 min, 83–77% A; 35–60 min, 77–53% A; 60–67 min, 53–86% A; 67–70 min, 86% A. The flow rate was 1 mL min^−1^, the monitored wavelength was 520 nm, and the column temperature was 35 °C. The anthocyanin concentrations were estimated according to 3-*O*-glucoside standards based on the method described by Fanali [57].

## 5. Conclusions

We cloned and characterized the *RdCHS1* gene from *R. delavayi* and demonstrated its role through in vitro and in vivo testing. We found that *RdCHS1* was expressed globally in all tissues and was not associated with anthocyanin synthesis during flower development, which revealed that *RdCHS1* may participate in the biosynthesis of not only anthocyanin, but also other flavonoids in *R. delavayi*. Meanwhile, an enzyme activity assay was conducted, which indicated that RdCHS1 possesses CHS activity, converting malonyl-CoA and *p*-coumaroyl-CoA substrates into naringenin chalcone. Furthermore, the physiological role of *RdCHS1* was studied in *Arabidopsis tt4* mutants and tobacco; the results showed that *RdCHS1* recovered the *tt4* mutant phenotypes and led to dark pink tobacco flowers, suggesting that the manipulation of *RdCHS1* may contribute to modifying the color of other ornamental plants.

## Figures and Tables

**Figure 1 molecules-29-01822-f001:**
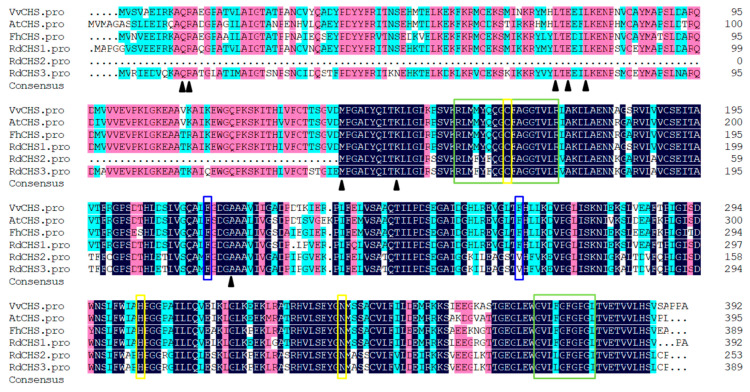
Amino acid sequence alignment of CHS protein in *R. delavayi* with proteins from other species. The yellow box represents three conserved catalytic residues in CHS. The blue-frame amino acids determine the specificity of the CHS substrate. The green rectangular box indicates the highly conserved domains of CHS. The black triangles represent important residues for binding to coumarinyl coenzyme A and the residues specific to the cyclic reaction of CHS.

**Figure 2 molecules-29-01822-f002:**
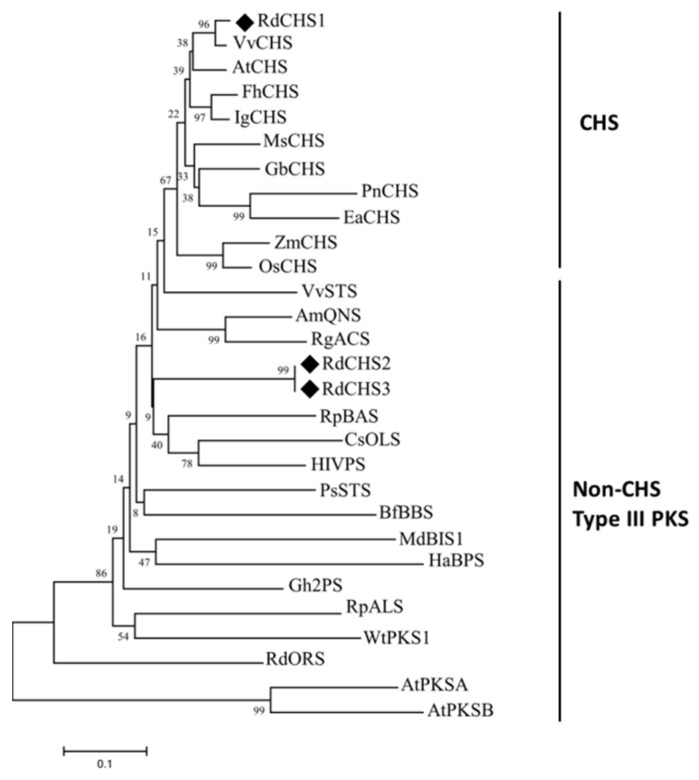
Phylogenetic analyses of RdCHSs. Plant species and GenBank accession numbers are as follows: FhCHS (*Freesia hybrida*, AEO45114.1), IgCHS (*Iris germanica*, BAE53636.1), VvCHS (*Vitis vinifera*, BAA31259.1), AtCHS (*Arabidopsis thaliana*, AAA32771.1), ZmCHS (*Zea mays*, CAA42763.1), OsCHS (*Oryza sativa*, BAA19186.2), GbCHS (*Ginkgo biloba*, AAT68477.1), MsCHS2 (*Medicago sativa*, P30074.1), PnCHS (*Psilotum rudum*, BAA87922), EaCHS (*Equisetum arvense*, Q9MBB1.1), AmQNS (*Aegle marmelos*, AGE44110), RgACS (*Ruta graveolens*, CAC14058.1), RpBAS (*Rheuam palmatum*, AAK82824.1), VvSTS (*Vitis vinifera*, ABV82966.1), PsSTS (*Pinus sylvestris*, CAA43165), Gh2PS (*Gerbera hybrida*, P48391.2), RpALS (*Rheum palmatum*, AAS87170), CsOLS (*Cannabis sativa*, B1Q2B6), HIVPS (*Huamulus lupulus*, ACD69659.1), HaBPS (*Hypericum androsaemum*, AAL79808.1), MdBIS1 (*Malus domestica*, NP001315967), BfBBS (*Bromheadia finiaysoniana*, CAA10514.1), WtPKS1 (*Wachendorfia thyrsiflora*, AAW50921), RdORS (*Rhododendron dauricum*, BAV83003), AtPKSA (*Arabidopsis thaliana*, O23674), and AtPKSB (*Arabidopsis thaliana*,Q8LDM2). QNS, OLS, ALS, BIS, and ORS stand for quinolone synthase, olivetol synthase, aloesone synthase, 3, 5-dihydroxybiphenol synthase, and orcinol synthase, respectively. Black diamond represents 3 *CHS* gene of *R. delavayi*.

**Figure 3 molecules-29-01822-f003:**
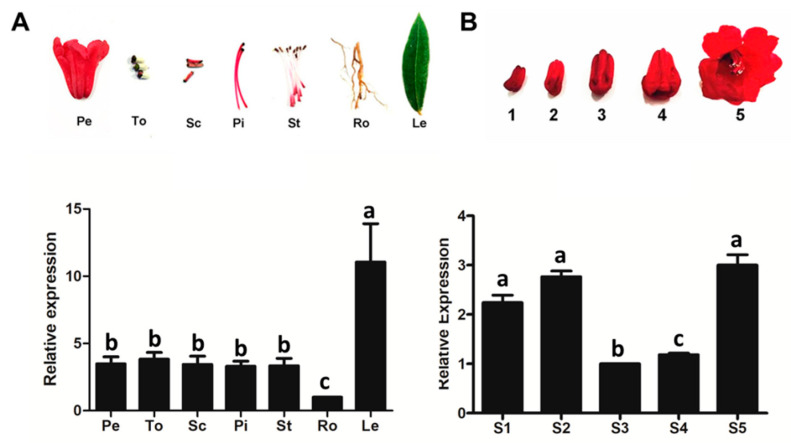
Expression profiles of RdCHS1 in R. delavayi. (**A**) Relative expression levels of RdCHS1 gene in different tissues; Pe, petals; To, toruses; Sc, scapes; Pi, pistils; St, stamens; Ro, roots; and Le, leaves. (**B**) Relative expression levels of RdCHS1 at five flower developmental stages: S1, flower buds about 1 cm; S2, flower buds about 1.5 cm; S3, flower buds about 2 cm; S4, freshly opened flowers; and S5, blooming flowers. Results represent means ± SE from three biological replicates. Letters a, b, c indicate very significant difference at the 0.01 level.

**Figure 4 molecules-29-01822-f004:**
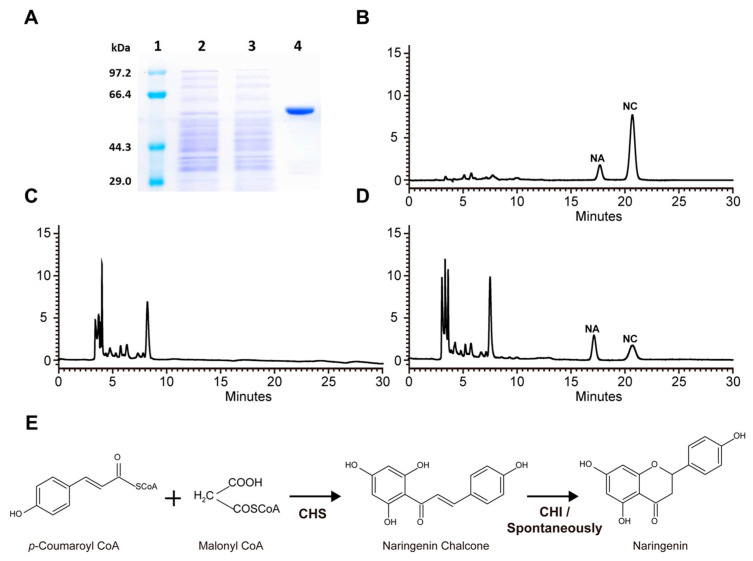
Biochemical assays of recombinant RdCHS1. (**A**) Expression of RdCHS1 in *E. coli*. (1) Maker; (2) total soluble protein from *E. coli* expressing pET-32a (+) vector; (3) total soluble protein from *E. coli* expressing *RdCHS1* prior to induction with IPTG; and (4) purified RdCHS1. (**B**) Standard of naringin chalcone; (**C**) the control (empty pET-32a vector); (**D**) HPLC profiles of the reaction products of RdCHS1; and (**E**) reaction scheme of the enzymatic synthesis of naringin chalcone/ naringin from malonyl-CoA and *p*-coumaroyl-CoA.

**Figure 5 molecules-29-01822-f005:**
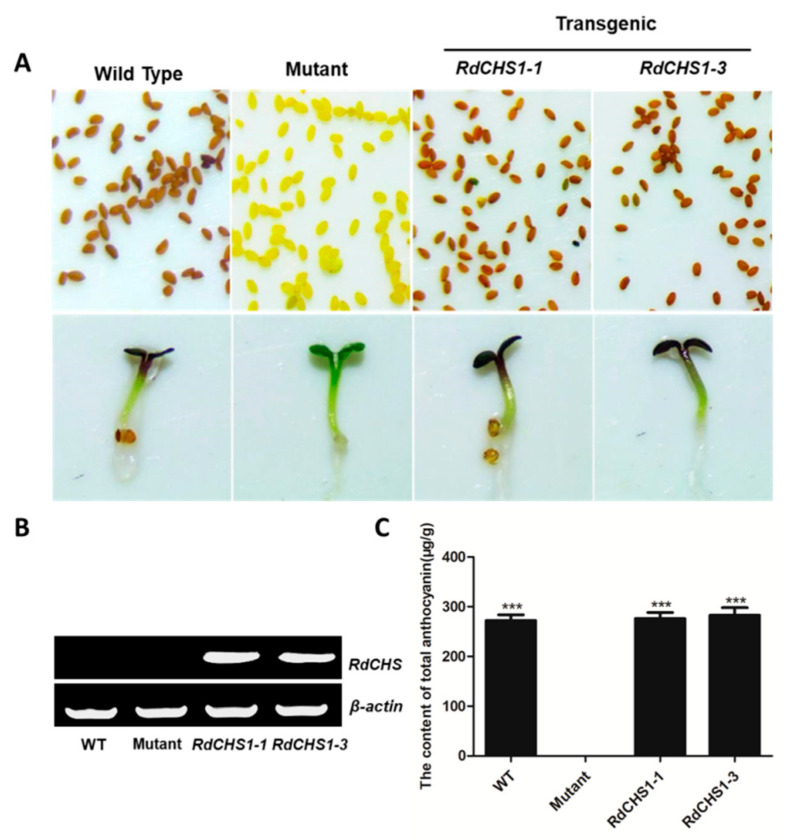
Complementation of RdCHS1 function in the Arabidopsis tt4 mutant. (**A**) Phenotype of the wild-type, mutant (*tt4*), and T2 transgenic lines. (**B**) Expressional analysis of RdCHS1 in wild-type, mutant, and transgenic lines. (**C**) Total contents of anthocyanins in Arabidopsis seedlings. The results correspond to the means from three biological replicates. The asterisks indicate the significant differences between the means of wild-type and transgenic plants calculated using Tukey’s HSD test (*** *p* < 0.001).

**Figure 6 molecules-29-01822-f006:**
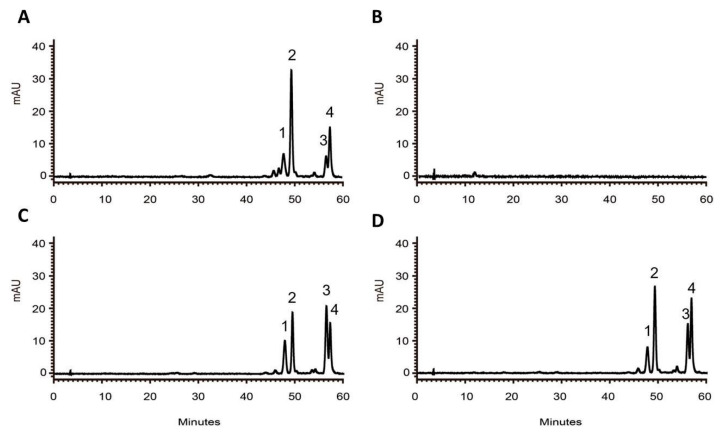
HPLC analyses of anthocyanins in Arabidopsis seedlings. HPLC chromatograms of the samples from the seedlings of wild-type (**A**), mutant (**B**), RdCHS1-1 (**C**), and RdCHS1-3 (**D**). Peaks 1–4 represent cyanidin 3-*O*-[2″-*O*-(xylosyl) 6″-*O*-(*p*-O-(glucosyl) *p*-coumaroyl) glucoside] 5-*O*-[6″″-*O*-(malonyl) glucoside, cyanidin 3-*O*-[2″-*O*-(6‴-*O*-(sinapoyl) xylosyl) 6″-*O*-(*p*-*O*-(glucosyl)-*p*-coumaroyl) glucoside] 5-*O*-(6″″-*O*-malonyl) glucoside, pelargonidin derivatives, and pelargonidin derivatives.

**Figure 7 molecules-29-01822-f007:**
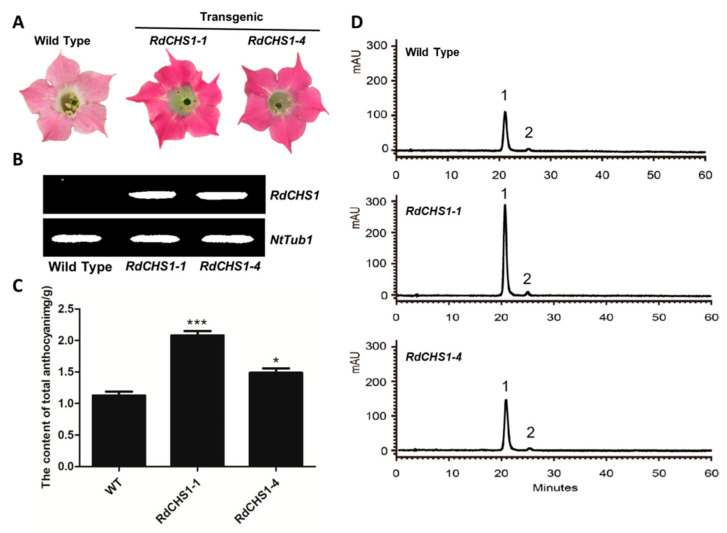
Effect of *RdCHS1* on anthocyanin accumulation in transgenic tobacco flowers. (**A**) Tobacco flowers of the wild-type and transgenic lines. (**B**) Expression profiles of *RdCHS1* in the flowers of transgenic tobacco. (**C**) Quantitation of anthocyanin accumulation levels in transgenic tobacco flowers with HPLC. (**D**) HPLC chromatograms of the samples from the flowers of wild-type and transgenic tobacco. The results correspond to means from three biological replicates. The asterisks indicate significant differences between the means of the wild-type and transgenic plants calculated using Tukey’s HSD test (*** *p* < 0.01; * *p* < 0.05).

**Figure 8 molecules-29-01822-f008:**
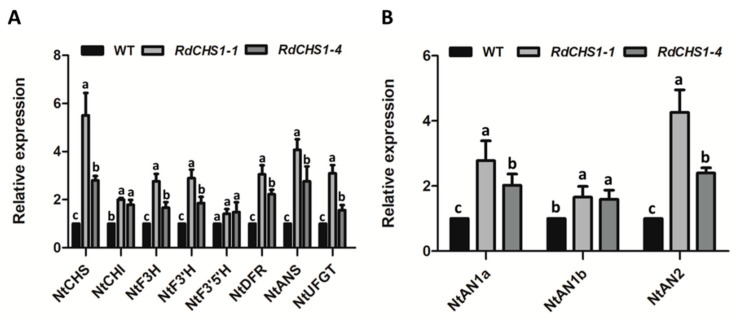
Expression analyses of endogenous anthocyanin biosynthetic genes in the corollas of transgenic tobacco. (**A**) The expression profiles of the structure genes in the corollas of transgenic tobacco. (**B**) Expression profiles of the regulatory genes in the corollas of transgenic tobacco. The results represent the means ± SE from three biological replicates. The different letters above the bars indicate significant differences between the samples judged through Tukey HSD tests (*p* < 0.01).

## Data Availability

Data are available within article and Appendix A.

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
