# Peer review of "Chalcone-Synthase-Encoding RdCHS1 Is Involved in Flavonoid Biosynthesis in Rhododendron delavayi"

_molecules, 2024, doi:10.3390/molecules29081822_

Round 1
Reviewer 1 Report
Comments and Suggestions for Authors
In their manuscript titled “Chalcone synthase-encoding RdCHS1 is involved in flavonoid biosynthesis in Rhododendron Delavayi,” Huang and colleagues report on their efforts to identify the enzyme(s) responsible for catalyzing the chalcone synthase step of anthocyanin biosynthesis in an ornamental flower. An existing transcriptional dataset was mined for genes encoding homologs to previously characterized CHS isoforms, with subsequent in silico analysis performed to narrow down the 3 candidates to a single presumed bonafide CHS. Through a combination of in vitro assays on recombinant RdCHS1 protein, complementation of the unpigmented phenotype of an Arabidopsis CHS mutant, and transgenic expression in tobacco, the authors demonstrate that RdCHS1 can catalyze the CHS reaction both in vitro and in vivo.
Overall, the manuscript presents compelling evidence that RdCHS1 can catalyze the CHS reaction and that this is sufficient to sustain production of flavonoids in plant systems. However, the key conclusion that RdCHS1 contibutes to flavonoid biosynthesis in R delavayi could be more strongly supported with some additional information and experimentation, as described in my specific comments below.
Main Comments:
1) RdCHS2 and RdCHS3 are not selected for experimental analysis due to amino acid substitutions at key residues. Some description of these key residues is given; however, is there prior evidence that the specific substitutions observed in RdCHS2 and 3 are incompatible with CHS function? If not, recombinant expression and enzyme assays could answer whether these two proteins can catalyze the CHS reaction.
2) Determining the tissue-specific expression profile of RdCHS2 and RdCHS3 may also provide insight about their possible function (either in addition to or in place of RdCHS1) in anthocyanin production.
3) Although enzyme assays were conducted showing that RdCHS1 can catalyze the CHS reaction in vitro, it would be quite useful to conduct more enzyme assays so as to determine kinetic parameters. This would be helpful because CHS-like enzymes have been reported to be promiscuous (as described in this manuscript), and kinetic parameters could help reveal weather the enzyme functions with the expected efficiency of a bonafide CHS, whereas low enzyme efficiency may indicate the CHS activity is a physiologically irrelevant side activity of a promiscuous enzyme.
4) The results in the Arabidopsis complementation experiment very clearly show that RdCHS1 can sustain anthocyanin production. However, it is not clear if this reflects physiological function in the host species, or if it’s an artifact of an artificial situation (potentially very high expression of an enzyme with potentially very poor catalytic efficiency). I recommend quantifying expression of RdCHS1 relative to the levels that AtCHS is normally expressed in Wild-type tissues, in addition to performing the kinetic characterization described in my prior point.
5) The finding that RdCHS1 expression induces higher expression of other pathway genes in tobacco is fascinating. However, it also calls into question whether the increased pigment production is due to catalytic function of RdCHS1 or if it’s an indirect effect due to the induction of the endogenous pathway. The manuscript should be revised to acknowledge this uncertainty.
6) Overall, the conclusions of endogenous function of RdCHS1 will remain somewhat speculative without testing directly in R delavayi. Are methods available to repress gene expression (perhaps through RNAi or VIGS) in this species?
Minor Points
7) There are numerous grammatical errors that at times made reading a little difficult, but which would be readily addressed through the use of a technical writing service.
8) The chromatograms have peaks labeled with numbers. While the corresponding compound names are listed with the numbers in supplemental files, it would be helpful to have these compound names directly in the figure legends.
9) It is not clear how the cloning of the gene into pET-32a and pBI121 vectors was performed, and therefore there is some uncertainty about the exact structure of the final constructs. I recommend providing more detail about the cloning methods used.
Comments on the Quality of English LanguageThere are numerous grammatical errors that at times made reading a little difficult, but which would be readily addressed through the use of a technical writing service.
Reviewer 2 Report
Comments and Suggestions for Authors
This manuscript reports cloning and functional characterization of the chalcone synthase gene from Rhododendron delavayi. Although many chalcone synthases have been characterized thus far, the current work represents a comprehensive study of such genes from Rhododendron delavayi. mRNA transcript analysis to monitor the expression of the chalcone synthase CHS1 gene is excellent as it convincingly reflects anthocyanin synthesis. Heterologous protein production and follow-up enzyme assays and complementation studies in tt4 mutant and transgenic tobacco unambiguously establish functional roles of chalcone synthase. The work presented in this manuscript broadens our understanding of such an important enzyme.
Minor comments
Page 1, line 40”…have been finished..”>>>have been completed
Page 2, line 53: .. snapdragon>>>> Antirrhinum sp
Page 2 line 56: For petunia, there are more than 8 copies of CHS genes in its 56 genome, but only CHS-A and CHS-J transcript in floral tissues and conspicuously 57 down-regulated in the white parts of ‘Red Star’>>> For petunia, there are more than 8 copies of CHS genes in its genome, but only CHS-A and CHS-J transcript are expressed (???) in floral tissues. They are conspicuously down-regulated in the white parts of ‘Red Star’
Author Response
Thank you and the reviewer 2 for the helpful suggestions and comments on our manuscript entitled "Chalcone synthase-encoding RdCHS1 is involved in flavonoid biosynthesis in Rhododendron Delavayi". During the past several days, we have worked extensively to address all the questions raised by the reviewers. Here are our point-to-point responses to the comments.
Reviewer 2:
Q1. Page 1, line 40”…have been finished..”>>>have been completed
Page 2, line 53: .. snapdragon>>>> Antirrhinum sp
Page 2 line 56: For petunia, there are more than 8 copies of CHS genes in its genome, but only CHS-A and CHS-J transcript in floral tissues and conspicuously down-regulated in the white parts of ‘Red Star’>>> For petunia, there are more than 8 copies of CHS genes in its genome, but only CHS-A and CHS-J transcript are expressed (???) in floral tissues. They are conspicuously down-regulated in the white parts of ‘Red Star’
Answer:
Thanks for your advice, we have revised above descriptions in the current version.
Thanks very much for your attention to our paper!
Best wishes for you!
Sincerely yours,
Wei Sun
2024/03/2